# The Contribution of Local Management to Biodiversity Conservation: An Analysis of Specific Cases in the Region of Madrid (Spain)

**Pedro Molina-Holgado [1], Nieves López-Estébanez [1], Ana-Belén Berrocal-Menárguez [2,*], Fernando Allende-Álvarez [1] and Miguel del Corro-Toro [3]**

[1] Department of Geography, Universidad Autónoma de Madrid, 28049 Madrid, Spain; pedro.molina@uam.es (P.M.-H.); nieves.lopez@uam.es (N.L.-E.); fernando.allende@uam.es (F.A.-Á.)

[2] Department of Transport Engineering, Urban and Regional Planning, ETSI Caminos, Canales y Puertos, Universidad Politécnica de Madrid, 28040 Madrid, Spain

[3] Tragsatec, Calle Maldonado 58, 28006 Madrid, Spain; mdct@tragsa.es

\* Correspondence: anabelen.berrocal@upm.es; Tel.: +34-910674197

**Abstract:** In line with the Urban Agenda for the EU, this article highlights the importance of local actions in the conservation of biodiversity, both through specific activities and by increasing the availability of information. As such, the policies and projects related to the conservation of biodiversity have been analyzed here at different levels and, in particular, the initiatives undertaken in the Madrid Region, Spain. Consequently, two cases are presented that demonstrate the role that local administrations can play in improving the biodiversity database, and hence, in the effective protection of areas of significant environmental value. First, we will examine the effects that creating an environmental inventory of vegetation, flora and landscape has had in Torrelodones. Second, among the more recent environmental policies implemented in the municipality of Madrid are those that resulted in the environmental recovery of the urban section of the Manzanares River. Both these actions demonstrate how local authorities can contribute to the conservation of biodiversity at relatively low expense and in line with EU guidelines. Notably, this occurred despite the fact that competences in environmental matters in Spain are not municipal. In this context, the paper reflects on the untapped potential of the *General Urban Planning Plans* (PGOU) in deep knowledge and sustainable and responsible management of municipal environmental values.

**Keywords:** biodiversity; public policies; local management; biodiversity database; renaturation plans

## 1. Introduction

Climate change and overexploitation of natural resources are causing a rapid loss of biodiversity in all of the planet's ecosystems [1,2]. The accelerated reduction in local populations of fauna and flora has a domino effect on the functioning of ecosystems and on human well-being [3,4]. In this context, the current health crisis due to the COVID-19 pandemic is further dramatic evidence that the risk of the emergence and spread of infectious diseases increases as biodiversity is destroyed [5,6]. In the present era, the changes to global environments are closely related to human action. The rate of decline in the abundance of species is so marked that the existence of a sixth mass extinction on the planet has been recognized. [7,8].

In Europe, land artificialization over the last two decades has been very dramatic, mainly as a result of urbanization. Maximum values for this process were recorded between 2000 and 2006, with an average of 1000 km²/year, and although there has been a downward trend since then, the average

value recorded by the European Environment Agency in the six-years between 2012 and 2018 was 539 km$^2$/year [9]. In the period 2000–2006, 23.5% of urban land in Europe was registered in Spain, situating the country as the principal nation in terms of land transformation, almost duplicating the numbers in the second country, France, with 12.2% of urban land [10].

Accelerated transformation of land and the associated loss of biodiversity is a growing concern worldwide, and particularly in Europe. Global agendas place particular emphasis on the need to conserve biodiversity due to the environmental benefits this provides. In this context, the main goal of the *United Nations Strategic Plan for Biodiversity 2011–2020* [11] is to address the underlying causes of biodiversity loss by taking biodiversity into consideration at all government and society levels. The *2030 Agenda for Sustainable Development* goes one step further, linking social well-being to the ecological vigor of associated ecosystems [12]. In Europe, one of the objectives of the European Biodiversity Strategy (EBS) for 2030 [13] is the effective protection of at least 30% of the marine and terrestrial areas. To this end, the EBS stresses the need to characterize and map spaces of environmental value in order to protect them, and to create ecological corridors within a true Trans-European Network for Natural Spaces. Special attention should be paid to the need to limit urban expansion, and to the preservation of the high diversity of the landscapes and land in agricultural, rural and peri-urban areas. Together, this strategy highlights the enormous interest, and the need to characterize and evaluate not only the areas that might contain elements of special interest but also, of all the spaces that are at risk of transformation due to their proximity to urban areas.

Accordingly, the creation of a national inventory of biodiversity in Spain was set out in the government legislation *7/2018* covering natural heritage and biodiversity ("*Ley 7/2018 de patrimonio natural y biodiversidad*"), a measure that was implemented through the *Royal Decree 556/2011 (20th April) for the Development of the Spanish Inventory of Natural Heritage and Biodiversity* ("*Real Decreto 556/2011, de 20 de abril, para el desarrollo del Inventario Español del Patrimonio Natural y la Biodiversidad*"). The creation and maintenance of this inventory is a task undertaken jointly by National and regional governments, exempting local entities from such responsibilities. This is due to the fact that in Spain, the competencies and remits related to the management and planning associated with natural resources are primarily the responsibility of the regional Autonomous Communities, such that local administrations fulfil only a residual role in this area that is essentially limited to the urban planning of their municipalities. These policies have a significant territorial impact, as they can determine the level of conservation or lead to the loss of areas of great ecological value [14]. Indeed, municipal urban planning has a strong impact on the conservation of biodiversity and local populations, often permitting changes in land use without a prior exhaustive assessment of the environmental status of spaces classified as land suitable for development. Although since 2006, in Spain the "General Urban Development Plans" ("Planes Generales de Ordenación Urbana" —PGOU) must be subjected to a "Strategic Environmental Assessment" ("Evaluación Ambiental Estratégica"—EAE) [15], as described in the legislation "*21/2013 on Environmental Assessment*" ("*Ley 21/2013 de Evaluación Ambiental*") [16], these assessments are usually based on the environmental inventories carried out on the regional level if such data exists. The poor quality and the lack of up-to-date information at the municipal level often results in the classification of spaces of environmental value as land that is suitable for development, which in some cases contain elements worthy of conservation and that may even be globally threatened or protected on a European level.

The environmental emergency that the planet is currently facing demands effective integration of conservation policies at various levels. A strategy must be adopted that allows the rapid and effective application of supranational policies at lower-ranking administrative and territorial levels, including at the municipal level. Moreover, the approach implemented must favor governance by local entities and citizen groups, as these can participate directly in maintaining and gathering information about local species and environments of high environmental value. These are the concepts proposed in the Urban Agenda for the European Union (EU), approved in 2016, which has a clear operational vocation [17]. The promotion of knowledge exchange between administrations is high among the

objectives of this document, proposing a new form of multilevel governance. Improvements in inter-institutional coordination and cooperation on urban issues will ensure that existing regulations with urban repercussions better reflect the real needs of cities and their administrations [18]. Indeed, the Urban Agenda for the EU specifically aims to foster the intervention of local administrations in the design of European policies in order to produce an impact on cities through the creation of partnerships, as well as to improve the knowledge base used to formulate urban planning policies, all without impinging on the current distribution of competences between administrations [19,20].

In line with the Urban Agenda for the EU, this article seeks to highlight the importance of local activities on the conservation of biodiversity, both through the development of specific actions and by enhancing the information available on which such actions can be based. To this end, the policies and programs associated with the conservation of biodiversity are analyzed at different levels, focusing in particular on initiatives undertaken in the Madrid Region (Spain). Consequently, two cases are presented that are representative of the role that local administrations can play in improving the biodiversity database, and hence, in the effective protection of areas of significant environmental value. First, we present the consequences of the creation of an environmental inventory of vegetation, flora and landscape in Torrelodones. Second, we present some of the latest environmental policies implemented in the municipality of Madrid that have resulted in the environmental recovery of the urban section of the Manzanares River.

These two cases make it clear how local actions implemented in line with the Urban Agenda for the EU can help redress the current crisis in biodiversity.

## 2. Materials and Methods

### 2.1. Study Area

The Mediterranean basin is considered a "hotspot" of great value in terms of its biodiversity [21], in particular the Iberian Peninsula where a close relationship has long existed between natural ecosystems and human activities [22], generating landscapes of great biological value [23,24]. It is precisely in this high-value space where the Autonomous Community of Madrid is located, the physical environment on which this study focuses, a territory of 8030.88 km$^2$ that housed 6,663,394 inhabitants in 2019 [25,26]. Despite its high land use rate (83,982 ha in 2005, 10.46% of the territory) [27], the region of Madrid is still an area of high biodiversity. This is largely due to the high concentration of the population in the metropolitan area, whereby 91.54% of the population resides in 33.5% of the regional territory (data from 2019) [28]. Other major factors include the minimal fragmentation of the territory and the existence of large extensions of public property, some larger than 15,000 ha, even in the municipal area of Madrid [29]. These include the Monte de El Pardo, a continuous holm oak grove of 15,821 ha located 7.8 km from the central point of the historic center of the city of Madrid [30], and integrated within the Special Area of Conservation ES3110004 in the EU Habitats Directive, the "Manzanares River Basin" ("Cuenca del Río Manzanares"), and in the Special Protection Area ES0000011 "Monte de El Pardo" under the EU Directive on the Conservation of Wild Birds [31].

The specific locations of the case studies that will be dealt with here are Madrid and Torrelodones, two municipalities with very different extensions and population sizes: Madrid (605.8 km$^2$), capital of the Region and of the State; and Torrelodones (21.9 km$^2$), a mid-sized municipality located to the north of Madrid (Figure 1). While the first of these municipalities, Madrid, has a population of 3,266,126 inhabitants, Torrelodones has a population of 23,714 inhabitants [32]. These values represent 49.02% and 0.27% of the regional population (6,663,394 inhabitants in 2019), respectively, and the municipalities constitute 7.55% and 0.27% of the surface area of the Autonomous Community (8033.8 km$^2$) [33]. Both are integrated into the Functional Zone of Madrid, the third largest Metropolitan Area in the European Union based on the number of inhabitants, following those of Paris and London [34,35].

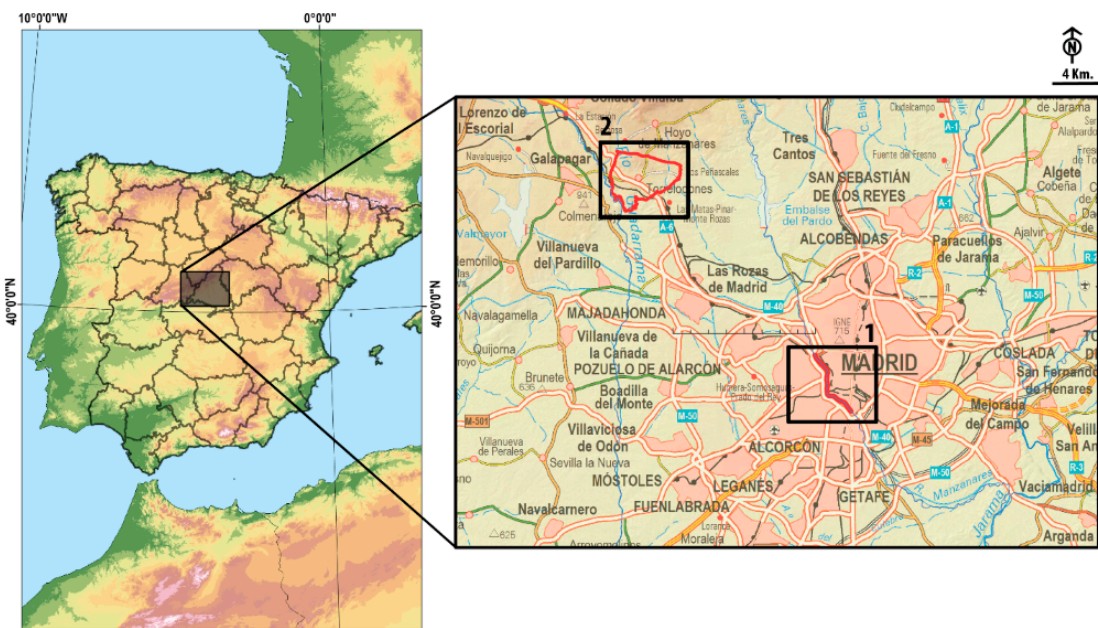

**Figure 1.** Location of the study area. Map base source: Spain DEM(Digital Elevation Model) from USGS (United States Geological Survey) Shuttle Radar Topography, 2004 [36]. Topographic info from the National Geographic Institute 1:200.000 [37].

## 2.2. Methods

In order to carry out this study, a thorough review of biodiversity policies and programs was carried out at four territorial levels for the period 2010–2019: the European, State, regional and local levels. At the highest level of detail, the documents produced by the 179 municipalities that are part of the region of Madrid were analyzed. Thus, information was obtained on the different programs, plans and actions that local entities have designed in relation to the conservation of biodiversity. A qualitative assessment of the impact and repercussion on conservation was carried out for each of the plans and projects analyzed, rating them from 1 (low) to 3 (high), according to the criteria indicated in Table 1. These criteria were defined by a panel of 3 experts, besides the authors of this papers, on the basis of their own professional and research experience.

**Table 1.** Criteria for the evaluation of local plans, projects and other initiatives.

|  | Rating | | |
|---|---|---|---|
|  | **Low (1)** | **Medium (2)** | **High (3)** |
| **Projects** | | | |
| Generic qualitative information | * | | |
| Comprehensive qualitative information | | | * |
| Quantitative information | | | * |
| Data accurate for the municipality | | * | |
| Data accuracy less than 1 km$^2$ UTM | | * | |
| 1 km$^2$ UTM data accuracy | | | * |
| Georeferenced data | | | * |
| **Plans, Programs and other initiatives** | | | |
| Quantifiable positive effects on biodiversity | | | * |
| Positive effects on biodiversity estimated but not quantified | | * | |
| No significant effects on local biodiversity | * | | |
| Agreements, conventions, monitoring bodies | * | | |

The examples of good local practices were analyzed distinctly for both of the cases as the interventions analyzed were not directly comparable: a restoration plan in the case of the activities designed by the Madrid City Council; a project to create and curate databases on local biodiversity in the case of the initiative implemented by the Torrelodones City Council. Nonetheless, both initiatives are described in detail.

The results and scope of the Plan for Renaturalization of the Manzanares as it passes through the city of Madrid ("Plan de renaturalización del río Manzanares a su paso por Madrid") have been assessed by considering its impact on the wintering populations of waterfowl and riparian birds. This required the collection of precise data, which was achieved by conducting a bi-weekly censuses in December 2019 and January 2020. Thus, data was obtained on the richness (number of species), abundance (number of individuals), the density of waterfowl (birds/km) and of riparian birds (birds 10 ha), and on their diversity, calculated in nats according to the Shanon index [38]. The wintering population of riparian birds was quantified based on the results obtained over four "census routes" [39], 25 + 25 m wide areas along the 6.8 km survey belts. The waterfowl data were global counts of the entire river section analyzed, also 6.8 km long, coinciding with the area affected by the renaturalization plan. These censuses are in line with the indications of the Field Protocol for waterbird counting [40].

The assessment of the initiative designed by the Torrelodones City Council, the creation of a database of local vascular flora biodiversity, examined its content in detail and evaluated the quality of the results obtained. The methodology used was examined and assessed, and the results were compared with those from other studies carried out both in the region of Madrid and beyond, considering its possible application at the local level. Most of the information included in this work is referenced by means of 1 km$^2$ UTM (Universal Transverse Mercator) grids (Figure 2) or by georeferencing, and the data was obtained during 599 h of field work carried out over 95 days, between April 2018 and September 2019. In this period, 238 visits were made to all the grids analyzed (mean 9.52, SD ± 5.3). The study also analyzed all the information regarding the local flora held in the databases of the *Iberian and Macaronesian Vegetation Information System* ("*Sistema de Información de la Vegetación Ibérica y Macaronésica*") [41], in the Biodiversity Data National Website ("Portal Nacional de Datos de Biodiversidad") [42] and in the *Spanish plant information system* ("*Sistema de Información sobre las plantas de España*") [43]. A preliminary catalogue of the municipality's flora was also taken into consideration [44], as were other local references from direct, unpublished sources, and data provided by some botanical blogs of proven credibility and value [45,46].

The data collected during the field work was processed with a Trimble Geo 7X differential GPS and integrated into ArcGis 10.5 geographic information system in shapefile format. This allowed us to generate the maps illustrated here. To calculate the density of certain species (ferns), kernel algorithms were employed to localize hot spots in their distribution areas. Data management was carried out using the Statgraphics Centurion 18 © and Microsoft Excel 2016 © software.

## 3. Results

### 3.1. Public Policies and Programs for the Protection of Biodiversity

After reviewing the available documentation and legislation on biodiversity and conservation, a total of 5 regulatory documents were found at a European level, 10 at a national level and 5 at a regional level (Table 2). On the European scale, the five documents that have the strongest impact are the directives regarding the protection of habitats and birds, and the recently updated EBS, together with the United Nations Convention on Biodiversity. All of them are very relevant to the maintenance of biodiversity and in cases, they are the essential element that has led to the designation of new levels of protection to spaces within the European territory in general, and in Spain and Madrid in particular. At the national level, three laws that have been passed were identified, two of which (*Law 7/2018, of July 20, amending Law 42/2007, of December 13, on natural heritage and biodiversity, and Law 45/2007, of December 13, for the sustainable development of the rural environment*) are of particular interest

owing to the importance and significance they have had at the regional and local levels. In addition, two strategies and one plan have been drawn up, as well as three royal decrees, which together have promoted the implementation of important actions that include the creation of a Natural Heritage and Biodiversity Inventory or the list of wild species under the special protection system, as well as the catalogue of threatened species.

On a regional level, this analysis produced very mixed results (Table 3). It is important to highlight the existence of the Forestry Plan for the Community of Madrid, as well as the Rural Development Program and the Catalogued Wetlands Action Plan. However, it is striking, to say the least, that the Regional Catalogue of Threatened Species of Wild Fauna and Flora dates back to 1992, since when it has not been updated.

Finally, at the local level (Table 4), only 11 of the 179 municipalities analyzed (6.2%) have carried out any activity to identify and conserve environmental values. In total, 20 projects were found that had different scopes and impacts in terms of conservation. Six of the projects were classified as low impact (rating 1), as they involved affiliations to biodiversity observatories and other projects that are carried out outside of the local entity. Six projects were considered to have a medium impact (rating 2): the study of the flora in Alpedrete; the biodiversity studies and the Catalogue in Aranjuez; the Catalogue of Biodiversity In Fuenlabrada; a microreserve pilot project in Madrid; or the biodiversity studies associated with the "Hayedo de Montejo" that although led by the Polytechnic University of Madrid, received municipal support (Montejo de la Sierra). Nevertheless, eight high-impact projects (rating 3) should be highlighted from the municipalities analyzed, the most relevant being: the study of the flora in Alpedrete; the biodiversity catalogue of Fuenlabrada; the Strategic Plan for Green Areas, Biodiversity and Trees in the Municipality of Rivas-Vaciamadrid; the Green Infrastructures and Biodiversity plan, and the plan for the renaturalization of the river Manzanares as it passes through the city of Madrid; as well as the plan for the management of Madrid's woodlands; and the connectivity strategy, together with the study of the local flora in Torrelodones.

**Table 2.** Conservation of biodiversity policies and projects on a global and European level relevant to the Autonomous Community of Madrid.

| Level | Legislation | | Year |
|-------|---|---|------|
| Global | - | EU Biodiversity Strategy for 2030 | 2020 |
| | - | United Nations Convention on Biological Diversity | 1992 |
| European | - | Council Directive 92/43/EEC of 21st May 1992 on the conservation of natural habitats, and of wild fauna and flora | 1992 |
| | - | Directive 2000/60/EC of the European Parliament and of the Council of 23rd October 2000 establishing a framework for Community action in the field of water policy | 2000 |
| | - | Spanish Forestry Plan, 5th July 2002 | 2002 |
| | - | Directive 2009/147/EC of the European Parliament and of the Council of 30th November 2009 on the conservation of wild birds | 2009 |

**Table 3.** Conservation of biodiversity policies and legislation at the national and regional level in the Autonomous Community of Madrid.

| Level | Law | Year |
|---|---|---|
| National | - Spanish strategy for the conservation and sustainable use of Biodiversity | 1993 |
| | - Law 3/1995, of 23 March, on Livestock Trails | 1995 |
| | - Law 45/2007, of 13 December, for the sustainable development of the rural environment | 2007 |
| | - Spanish national strategy for sustainable development | 2007 |
| | - Royal Decree 556/2011, of 20th April, for the Development of the Spanish Inventory of Natural Heritage and Biodiversity | 2011 |
| | - Royal Decree 139/2011, of 4th February, for the development of a List of Wild Species under the Special Protection System and of the Threatened Species Catalogue | 2011 |
| | - Royal Decree 216/2019, of 29th March, which approves the list of invasive alien species of concern for the outermost region of the Canary Islands, amending Royal Decree 630/2013, of August 2nd, which regulates the Spanish catalogue of invasive alien species | 2013 |
| | - Law 7/2018, of 20 July, amending Law 42/2007, of December 13th, on natural heritage and biodiversity | 2018 |
| Regional | - Decree 18/1992, of 26 March, which approves the Regional Catalogue of threatened species of wild fauna and flora, and establishes the unique trees category | 1992 |
| | - Decree 50/1999, of 8 April, which approves the Forestry Plan of the Community of Madrid | 1999 |
| | - Order 3382/2007, of 31 December, of the Regional Ministry of Environmental and Territorial Planning, by which the regulatory bases are approved for subsidies to local entities for the implementation and adaptation of the urban trees conservation plans and the urban trees inventory, as required by law 8/2005 for the protection and promotion of urban trees in the Community of Madrid | 2007 |
| | - 2014–2020 Rural Development Program in the Community of Madrid | 2014 |
| | - Decree 26/2020, of 8 April, of the Governing Council, which approves the Catalogued Wetlands Action Plan in the Autonomous Community of Madrid | 2020 |

**Table 4.** Conservation of biodiversity policies and legislation at the local level in the Autonomous Community of Madrid.

| Municipality | | Plan/Project | Rating |
|---|---|---|---|
| Alpedrete | - | Framework agreement governing the collaboration between the Alpedrete City Council and the Spanish Ornithological Society SEO/BIRDLIFE | 1 |
| | - | The flora of Alpedrete: Trees and forests, shrub communities and herbaceous species | 2 |
| Aranjuez | - | White paper on the biodiversity and conservation of the natural heritage in Aranjuez | 2 |
| | - | Catalogue of the trees and singular arboreal assemblages in Aranjuez | 2 |
| | - | LIFE ELM Project—Aranjuez City Council/Polytechnic University of Madrid | 1 |
| Fuenlabrada | - | Fuenlabrada biodiversity catalogue | 2 |
| Hoyo de Manzanares | - | Biodiversity observatory | 1 |
| Madrid | - | Green Infrastructure and Biodiversity Plan | 3 |
| | - | Plan for managing the Madrid woodlands | 3 |
| | - | Plan for the renaturalization of the river Manzanares as it passes through the city of Madrid | 3 |
| | - | Microreserve pilot project in the Empress Maria de Austria park | 2 |
| Miraflores de la Sierra | - | Agreement with the virtual biodiversity platform to create a citizen's biodiversity observatory | 1 |
| Montejo de la Sierra | - | Collaboration agreement for the studies of the "Hayedo de Montejo de la Sierra" | 2 |
| Rivas-Vaciamadrid | - | Strategic Plan for Green Zones, Biodiversity and Trees in the municipality of Rivas-Vaciamadrid | 3 |
| Soto del Real | - | Citizen's biodiversity observatory | 1 |
| | - | Study and proposals for interventions on livestock trails and public roads in Soto del Real (Madrid) | 3 |
| Torrelodones | - | Connectivity strategy | 3 |
| | - | Vegetation and flora in Torrelodones | 3 |
| | - | Torrelodones environmental guide: vegetation, flora and landscapes | 3 |
| Villalbilla | - | Forests without Borders within the "Friends of Trees" Network of Municipalities | 1 |

One of the most relevant initiatives at the local level is perhaps that developed as part of the "Agenda 21" action plan, prompted at the Earth Summit held in Rio de Janeiro in 1992. As part of this plan, local entities take leadership and organize sustainable development. At present, the municipal "Agenda 21" programs in the Madrid region are relatively poorly developed or have not yet started, and many have fallen by the wayside. However, we highlight the Local Agenda 21 of the Madrid City Council, carried out with the participation of district committees and through which sustainability diagnoses have been made in specific districts [47].

*3.2. Some Examples of Good Practices at the Local Level: The Cases of Madrid and Torrelodones*

Among all the programs and projects carried out by the local administrations in the Autonomous Community of Madrid, special mention should be made of the initiatives implemented in the

municipalities of Torrelodones and by the Madrid City Council. The results of the initiatives promoted by the local Council in Torrelodones are a good example of the important role that local administrations can play in improving biodiversity databases. The plan implemented by the Madrid City Council reveals the value of local intervention projects in restoring degraded areas, in the protection of biodiversity and in encouraging public usage. It is important to note that both are low-budget projects, costing €6000 and €1,216,054, respectively, part of public investment programs designed and implemented within a framework of social and environmental returns [48].

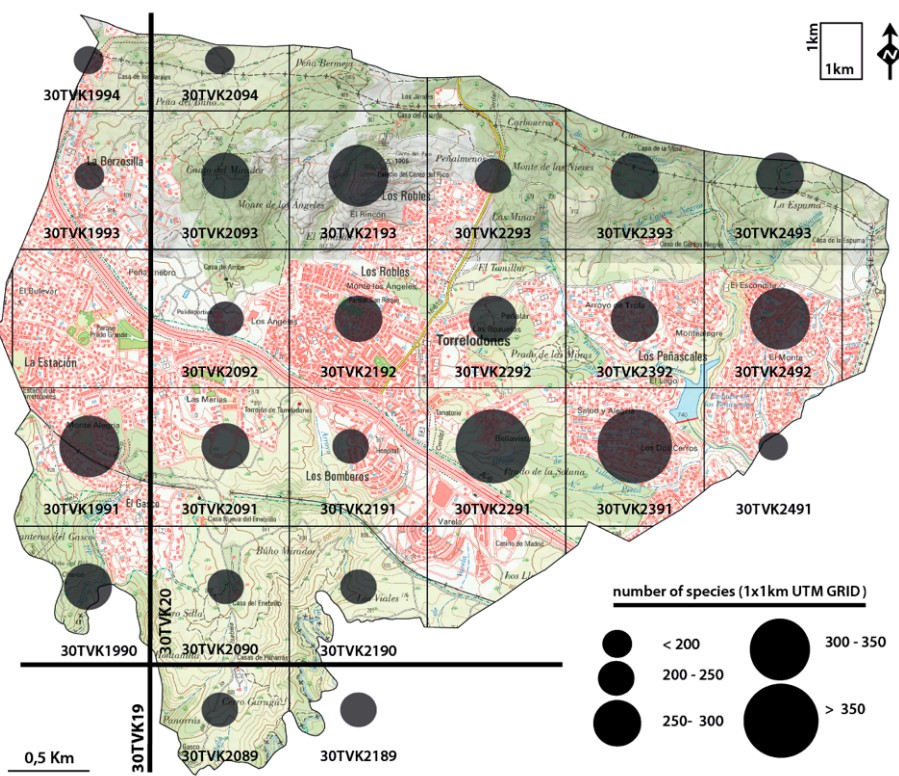

**Figure 2.** Number of species per 1 km$^2$ UTM (Universal Transverse Mercator) grids in the municipality of Torrelodones (Madrid). Data Source: [49].

### 3.2.1. The Initiative in the Municipality of Torrelodones, an Example of the Importance of Local Action for the Creation of Biodiversity Databases

The Study of the flora and vegetation of the municipality of Torrelodones, indicating the territories and areas of special interest, and a proposal of measures to conserve them is perhaps the most interesting initiative developed in this municipality to date. Some results from this study were published [49], although a significant amount of information remains unpublished [50–52] or is still being adequately prepared. These and others studies being carried out are derived, to a large extent, from the major lines of action identified in the 2013–2025 Torrelodones Participatory Strategic Plan [53]. This plan was designed to specify the socioeconomic and territorial development of the municipality, taking into account the opinion of the social and economic stakeholders, and of the general population.

The objectives of the project "*A study of the flora and vegetation of the municipality of Torrelodones*" were: to describe the flora and vegetation of the municipality of Torrelodones in detail; to characterize, map and assess territories and areas of special interest, identifying any potential risks to these; to locate elements of special interest due to their population dynamics or state of conservation; to propose measures for the management and conservation of vegetation in general, and of sensitive elements in particular. The information that was gleaned has resulted in the production of the following technical-scientific documents: a catalogue of the vascular flora of Torrelodones; an atlas of the local

flora per 1 km² UTM grids; a georeferenced database of unique flora and fauna in Torrelodones; the identification of areas and territories of special interest, and proposed measures to manage these.

The resulting catalogue of flora includes 651 taxa that were identified through the field work undertaken in 2018–2019, as well as another 89 taxa that were not observed during this field work but that were cited in the sources consulted. An additional 15 taxa were also incorporated, which are referred to in an addendum that includes data from September to December 2019, and another 12 were referred to in a later addendum with data from 2020 [50]. The number of visits per grid was strongly and positively correlated with the richness of the data (number of taxa detected ($r_s = 0.85$, $p < 0.001$ $n = 25$: $y = 96.368$ $ln$ $(x) + 47.936$; $R^2 = 0.753$)). Therefore, 767 taxa were identified in total, with statistically significant differences in richness across the 25 grids surveyed ($k = 177.93$, $p < 0.005$; $n = 25$: Figure 2).

It is important to note that this project situated 767 species in the municipal area of Torrelodones (0.27% of the region's surface area), a value that represents 29.16% of the regional flora (a total of 2630 species) [54]. Moreover, 2 of the 22 genus that are endemic to the Iberian Peninsula were among those identified, both monospecific: *Ortegia* Loefl. ex L. (*Caryophyllaceae*); and *Pterocephalidium* G. López (*Dipsacaceae*) [55]. In addition, the efforts to localize all the taxa in 1 km² UTM grids provided semi-quantitative information on the abundance of these species in each grid (Table 5), as well as on their global distribution in the area under study (Table 6).

**Table 5.** Abundance per 1 km² UTM grid.

| Extremely Rare [1] | Very Rare [2] | Rare [3] | Uncommon-Common but Localized [4] | Common [5] | Very Common [6] |
|---|---|---|---|---|---|
| Taxa with only one record/grid. | Elements sparsely distributed, present in low numbers (<5 records/grid) and widely dispersed | Infrequent elements (5–10 records/grid), that are sparsely distributed and present in low numbers across most of the study area | Infrequent elements (10–20 records/grid) or elements present in a few territories, yet abundant in those locations | Elements distributed widely over most of the study area (21–50 records/grid) and frequently present | Elements distributed widely over the study area (>50 records/grid) and very frequently present |

**Table 6.** Global distribution of the species.

| Localized | Narrow Distribution | Medium Distribution | Wide Distribution | Entire Territory |
|---|---|---|---|---|
| Taxon present in less than 20% of the grids surveyed | Taxon present in 21–40% of the grids surveyed | Taxon present in 41–60% of the grids surveyed | Taxon present in 61–80% of the grids surveyed | Taxon present in over 80% of the grids surveyed |

The work also incorporated a georeferenced database of singular species from which general information was derived, such as that shown in Figure 3. To date, this database includes 395 elements belonging to 30 rare or locally threatened species, among others: *Quercus suber* L., *Quercus faginea* Lam. subsp. *faginea*, *Arbutus unedo* L., *Acer monspessulanum* L., *Erica arborea* L. or *Lathyrus clymenum* L. In addition, the study localized the areas and territories of greatest botanical interest, proposing generic management measures, also identifying all the Habitats in Annex I of the Habitat Directive. Accordingly, 18 types of habitats were identified (Table A1), 11 of which were not listed in the *Spanish Inventory of Terrestrial Habitats*, one of the basic tools for the creation of the "Natura 2000" Network in Spain [56]. Notably, 3 of the 11 new habitats have a priority status since, and as indicated in the Directive itself, the "*priority natural habitat types means natural habitat types in danger of disappearance, which are present on the territory referred to in Article 2 and for the conservation of which the Community has particular responsibility in view of the proportion of their natural range ( . . . )*" [57].

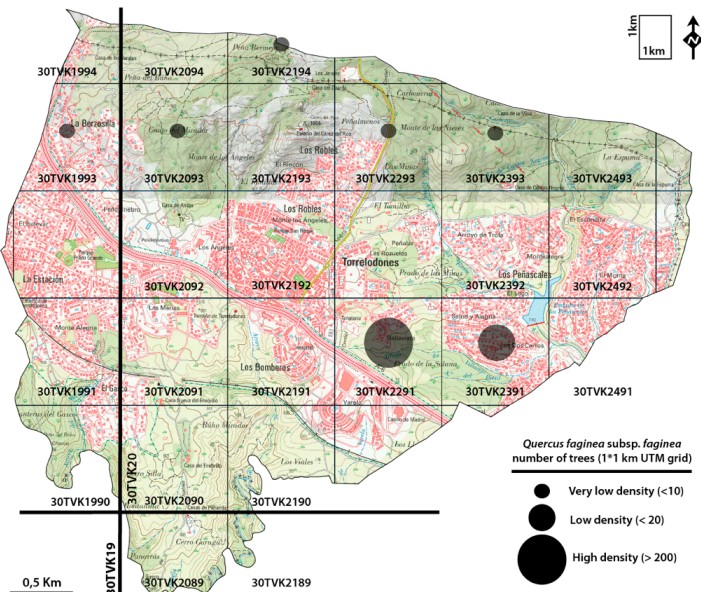

**Figure 3.** Distribution of *Quercus faginea* Lam. subps. *faginea* (density of trees per 1 × 1 km mapping grid). Map Base Source: Topographic info from National Geographic Institute 1:25,000 [58]. Data Source: [49,50].

In a later stage of the project, territories and areas susceptible to become flora microreserves were identified (Figure 4), and an analyses and quantification of the pteridoflora was performed. In this regard, studies carried out in 2019–2020 that also affect a neighboring municipality [50–52] provided quantitative data from 7857 elements belonging to 17 species of ferns, detected in 24 UTM 1 km$^2$ grids, as well as georeferenced data of 2391 elements belonging to 13 species.

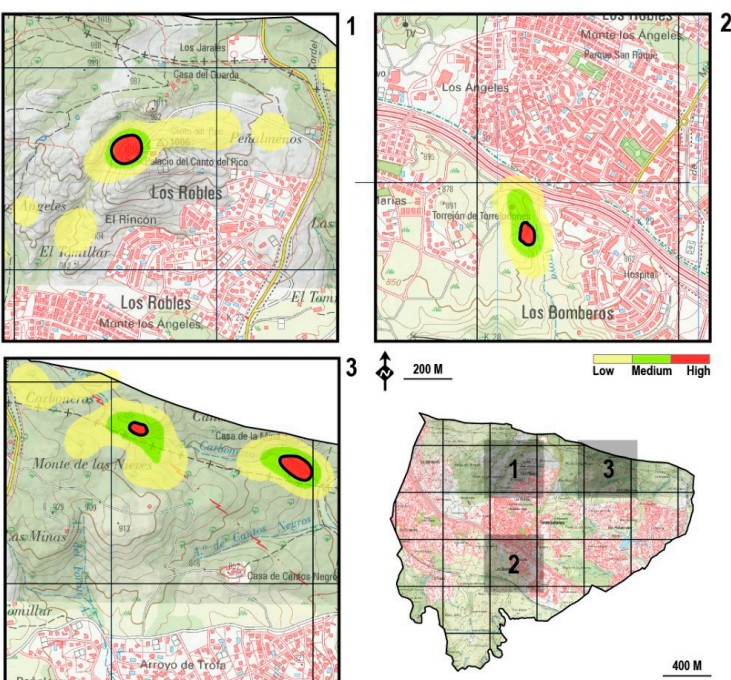

**Figure 4.** Torrelodones Potential areas of interest for ferns (density per 1 × 1 km mapping grid). 1. Canto del Pico; 2. Torreón; 3. Carboneros-Cancho de las Cruces. Map Base Source: Topographic info from National Geographic Institute 1:25,000 [58]. Data Source: [51,52].

It is worth noting that 78 nuclei of *Ophioglossum lusitanicum* L. were identified (30TVK2096-30TVK2097, 30TVK2194-2196, 30TVK2294, 30TVK2296, 30TVK2393), a species that has only been referred to once previously in Madrid, nearly 40 years ago [59,60], and there were 10 nuclei of *Ophioglossum azoricum* C. Presl. (30TVK2096, 30TVK2196, 30TVK2296), an element for which only 4 nuclei have been previously been described in the region [61]. A population of *Paragymnopteris marantae* (L.) K.H. Shing. subsp. *maranthae* (syn. *Notholaena marantae* L.) was also detected (30TVK2393) [51], a species whose presence has not previously recorded in the region of Madrid [62] and that is of great interest in terms of conservation, the closest populations of which are situated more than 115 km south-southwest [63] (Figure 5). Furthermore, the populations of a considerable number of species that are rarely or sparse distributed were geolocated and quantified in the region, including: *Allosorus pteridioides* (Reichard) Christenh. (syn. *Cheilanthes maderensis* Lowe, 1 nucleus; *Anogramma leptophylla* (L.) Link., 883 nuclei; *Asplenium adiantum-nigrum* L. var. *Adiantum-nigrum*, 147 nuclei; *Asplenium septentrionale* (L.) Hoffm. subsp. *septentrionale*, 462 nuclei; *Cystopteris dickieana* R. Simp., 508 nuclei; or *Polypodium interjectum* Shivas, 36 nuclei.

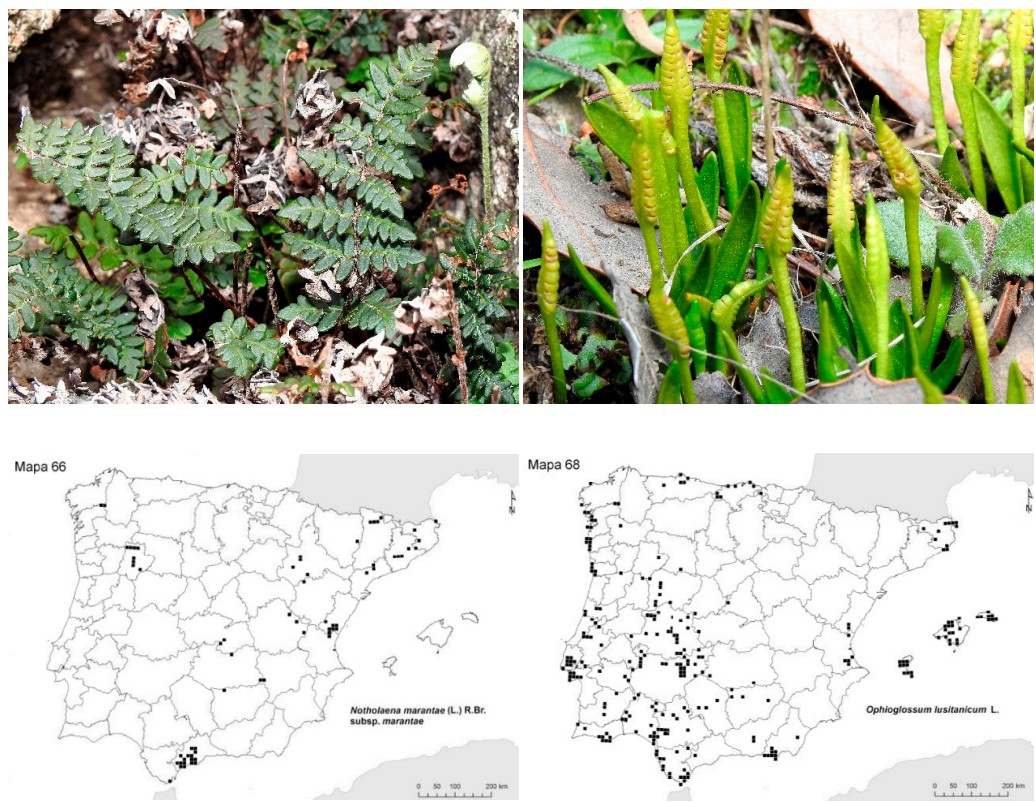

**Figure 5. Top left**, *Paragymnopteris marantae* (L.) K.H. Shing subsp. *marantae* (habitat of the species). **Top right**, *Ophioglossum lusitanicum* L. (habitat of the species and detail of the fertile segment). Bottom, distribution of *Paragymnopteris marantae* (L.) in the Iberian Peninsula per $10 \times 10$ km$^2$ UTM grids (**left**) and *Ophioglossum lusitanicum* L. (**right**) (14.2.2020). Images source: [51]. Maps source: [62].

### 3.2.2. Plan for the Renaturalization of the River Manzanares (Madrid), an Action of Importance for the Conservation of Regional Biodiversity

The renaturalization of the River Manzanares is probably the most beneficial activity for biodiversity carried out in the city of Madrid to date. The project began in 2016 as part of the *Plan for the renaturalization of the River Manzanares as it passes through the city of Madrid*, based on the proposal by the Non-Governmental Organization *Ecologists in Action-Madrid*. This initiative was undertaken and implemented by the Madrid City Council between 2015 and 2019 [64]. The primary goal of this project was to recover the function of the river as ecological corridor as it passes through

the city, connecting the upstream and downstream river sections. An improvement of the diversity of the urban fluvial space was also planned, in line with that set out in article 1.1 of the *Water Framework Directive* [65] and in the first objective of the Spanish National Strategy for River Restoration: "*To promote the integration of river ecosystems management into land-use and management policies, with criteria of sustainability*" [66,67].

This plan was implemented from May 2016 to May 2019, and it included the following actions along an urban river section of 7.5 km [68]:

- Activities involving the hydrological regime: opening of all the gates that regulate the water flow along the urban section of the river;
- Activities affecting the morphology of the channel and its bank, including the partial removal of the breakwaters at each end of the section and the conditioning of its embankment, creating deflectors using bioengineering techniques to establish a meandering river course in the central section;
- Activities involving the vegetation and fauna;
- Revegetation of the breakwaters by planting autochthonous species of riparian trees and shrubs (more than 5000 specimens) (*Alnus glutinosa, Crataegus monogyna, Fraxinus angustifolia, Populus alba, Rosa canina, Salix alba, Salix atrocinerea, Salix fragilis, Salix salviifolia, Sambucus nigra, Tamarix gallica, Ulmus minor*), eliminating exotic species and installing nesting boxes and releasing stripe-necked terrapins (*Mauremys leprosa*).

The success of the restoration of the River Manzanares has been remarkable. The recovery of this river section is particularly significant for birds during winter (December 2019–January 2020), a period in which 51 species of birds, 15 species of waterfowls and 36 species of riparian birds were recorded in the four fortnights analyzed. The mean global figures for waterfowl ranged between 1419–1951 birds (average 1690.50 ± 253.53: Table 7), with no statistically significant differences between the data from the 4 censuses carried out (*k = 0.5403; p = 0.9099; n = 15*). The density of the waterfowls reached an average value of 246.68 ± 37.21 birds/km, a significant figure considering that prior to the renaturalization of the river, only 5 species regularly used this fluvial space and at a very low density (<20 birds/km). The density of the three dominant species (*Anas platyrhynchos, Gallinula chloropus, Chroicocephalus ridibundus*) represents 89.09% of the global density, which means the values of diversity are low (1.23 ± 0.11 nats), although it must be borne in mind that it is an urban river section whose recovery began in 2016 and ended in May 2019.

**Table 7.** The Abundance (A), Kilometric Index of Abundance (A: birds/km), Richness (global values) and Diversity (nats) of Waterfowl Communities of the river Manzanares in Madrid.

| | 5 December 2019 | | 23 December 2019 | | 14 January 2020 | | 27 January 2020 | | | |
|---|---|---|---|---|---|---|---|---|---|---|
| **Species** | **A** | **KIA** | **A** | **KIA** | **A** | **KIA** | **A** | **KIA** | **Average** | **SD** |
| *Actitis hypoleucos* | 0 | 0.00 | 1 | 0.15 | 0 | 0.00 | 0 | 0.00 | 0.04 | * |
| *Alcedo atthis* | 6 | 0.87 | 4 | 0.58 | 2 | 0.29 | 6 | 0.87 | 0.66 | 0.28 |
| *Alopochen aegyptiaca* | 18 | 2.62 | 16 | 2.33 | 12 | 1.75 | 13 | 1.90 | 2.15 | 0.40 |
| *Anas platyrhynchos* | 273 | 39.80 | 277 | 40.38 | 217 | 31.63 | 309 | 45.04 | 39.21 | 5.57 |
| *Ardea cinerea* | 2 | 0.29 | 1 | 0.15 | 0 | 0.00 | 0 | 0.00 | 0.11 | * |
| *Cairina moschata* | 1 | 0.15 | 1 | 0.15 | 1 | 0.15 | 1 | 0.15 | 0.15 | 0.00 |
| *Chroicocephalus ridibundus* | 1082 | 157.73 | 890 | 129.74 | 875 | 127.55 | 715 | 104.23 | 129.81 | 21.90 |
| *Ciconia ciconia* | 4 | 0.58 | 0 | 0.00 | 0 | 0.00 | 0 | 0.00 | 0.15 | * |
| *Egretta garzetta* | 10 | 1.46 | 3 | 0.44 | 11 | 1.60 | 8 | 1.17 | 1.17 | 0.52 |
| *Gallinago gallinago* | 2 | 0.29 | 1 | 0.15 | 1 | 0.15 | 2 | 0.29 | 0.22 | 0.08 |
| *Gallinula chloropus* | 132 | 19.24 | 102 | 14.87 | 147 | 21.43 | 191 | 27.84 | 20.85 | 5.40 |

**Table 7.** *Cont.*

| Species | 5 December 2019 | | 23 December 2019 | | 14 January 2020 | | 27 January 2020 | | Average | SD |
|---|---|---|---|---|---|---|---|---|---|---|
| | A | KIA | A | KIA | A | KIA | A | KIA | | |
| *Larus fuscus* | 315 | 45.92 | 118 | 17.20 | 259 | 37.76 | 702 | 102.33 | 50.80 | 36.42 |
| *Lymnocryptes minimus* | 2 | 0.29 | 2 | 0.29 | 2 | 0.29 | 2 | 0.29 | 0.29 | 0.00 |
| *Phalacrocorax carbo* | 9 | 1.31 | 3 | 0.44 | 9 | 1.31 | 8 | 1.17 | 1.06 | 0.42 |
| *Tachybaptus ruficollis* | 0 | 0.00 | 1 | 0.15 | 0 | 0.00 | 0 | 0.00 | 0.04 | * |
| Abundance/KIA | 1856 | 270.55 | 1419 | 207.00 | 1536 | 223.91 | 1951 | 285.28 | 246.68 | 37.21 |
| Richness (global values) | 13 | | 15 | | 13 | | 11 | | 13 | 1.41 |
| Diversity (nats) | 1.24 | | 1.07 | | 1.20 | | 1.39 | | 1.23 | 0.11 |

\* Standard deviation (SD) higher than the average values ($\overline{X}$).

There were no statistically significant differences in the density values of riparian birds in the period analyzed ($k = 1.3359$; $p = 0.7206$; $n = 36$), with the global value of the surveyed section ranging from 86.86–178.08 birds/10 ha (average 143.47 ± 40.17). The generalist species linked to urban environments (*Columba livia domestica*, *Passer montanus*, *Pica pica*) were the most common elements (143.47 ± 40.17 birds/10 ha), the density of which represents 54.80% of the total density. Nonetheless, the density values of some species that were rare or absent prior to carrying out the restoration work were also relatively high, such as *Cettia cetti* (4.05 ± 1.63; 2.84%), *Passer montanus* (8.09 ± 5.10 birds/10 ha; 5.64%), *Motacilla alba* (5.90 ± 1.45 birds/10 ha; 4.11%) or *Ptyonoprogne rupestris* (2.99 birds/10 ha; 2.08%: Table 8). The sighting of the first two elements before the implementation of the recovery plan was accidental, and the presence of *Passer montanus* and *Ptyonoprogne rupestris* was particularly noteworthy due to their density and state of conservation. At present, the density of Sparrow tree in the study area is higher than that observed in the best areas of the Iberian Peninsula [69]. Moreover, the presence of Crag martin is also relevant, since it is an element in *Moderate decline* during wintering all over Spain (period 2008/2009, 2016, 2018) [70].

The mean value of the diversity of riparian birds communities was also moderately high (2.35 ± 0.30 nats), despite the area being only recently restored and with the vegetation still slightly under developed (Figure 6).

**Table 8.** The Abundance (A), Density (D) (birds/10 ha), Richness (global values) and Diversity (nats) of Riparian Bird Communities in the river Manzanares in Madrid.

| Species | 5 December 2019 | | 23 December 2019 | | 14 January 2020 | | 27 January 2020 | | Average | SD |
|---|---|---|---|---|---|---|---|---|---|---|
| | A | D | A | D | A | D | A | D | | |
| *Aegithalos caudatus* | 2 | 0.58 | 12 | 3.50 | 8 | 2.33 | 1 | 0.29 | 1.68 | 1.51 |
| *Carduelis carduelis* | 11 | 3.21 | 0 | 0.00 | 0 | 0.00 | 2 | 0.58 | 0.95 | * |
| *Chloris chloris* | 0 | 0.00 | 0 | 0.00 | 2 | 0.58 | 1 | 0.29 | 0.22 | * |
| *Spinus spinus* | 2 | 0.58 | 10 | 2.91 | 0 | 0.00 | 1 | 0.29 | 0.95 | * |
| *Certhia brachydactyla* | 3 | 0.87 | 4 | 1.17 | 2 | 0.58 | 1 | 0.29 | 0.73 | 0.38 |
| *Cettia cetti* | 16 | 4.66 | 19 | 5.54 | 15 | 4.37 | 6 | 1.75 | 4.08 | 1.63 |
| *Columba livia* | 146 | 42.55 | 210 | 61.21 | 225 | 65.58 | 72 | 20.99 | 47.58 | 20.35 |
| *Columba palumbus* | 4 | 1.17 | 3 | 0.87 | 7 | 2.04 | 22 | 6.41 | 2.62 | 2.57 |
| *Corvus monedula* | 2 | 0.58 | 0 | 0.00 | 2 | 0.58 | 0 | 0.00 | 0.29 | * |
| *Cyanistes caeruleus* | 11 | 3.21 | 10 | 2.91 | 8 | 2.33 | 6 | 1.75 | 2.55 | 0.65 |
| *Dendrocopos major* | 3 | 0.87 | 0 | 0.00 | 0 | 0.00 | 0 | 0.00 | 0.22 | * |
| *Dryobates minor* | 0 | 0.00 | 5 | 1.46 | 0 | 0.00 | 0 | 0.00 | 0.36 | * |
| *Emberiza schoenichlus* | 1 | 0.29 | 1 | 0.29 | 0 | 0.00 | 0 | 0.00 | 0.15 | * |
| *Erithacus rubecula* | 7 | 2.04 | 18 | 5.25 | 18 | 5.25 | 10 | 2.91 | 3.86 | 1.64 |
| *Fringilla coelebs* | 2 | 0.58 | 2 | 0.58 | 2 | 0.58 | 6 | 1.75 | 0.87 | 0.58 |
| *Motacilla alba* | 24 | 7.00 | 25 | 7.29 | 15 | 4.37 | 17 | 4.95 | 5.90 | 1.45 |

Table 8. *Cont*.

| Species | 5 December 2019 | | 23 December 2019 | | 14 January 2020 | | 27 January 2020 | | Average | SD |
|---|---|---|---|---|---|---|---|---|---|---|
| | **A** | **D** | **A** | **D** | **A** | **D** | **A** | **D** | | |
| *Motacilla cinerea* | 9 | 2.62 | 7 | 2.04 | 3 | 0.87 | 5 | 1.46 | 1.75 | 0.75 |
| *Myiopsitta monachus* | 10 | 2.91 | 10 | 2.91 | 17 | 4.95 | 5 | 1.46 | 3.06 | 1.44 |
| *Parus major* | 2 | 0.58 | 4 | 1.17 | 2 | 0.58 | 5 | 1.46 | 0.95 | 0.44 |
| *Passer domesticus* | 68 | 19.82 | 115 | 33.52 | 91 | 26.52 | 62 | 18.07 | 24.48 | 7.04 |
| *Passer montanus* | 53 | 15.45 | 15 | 4.37 | 26 | 7.58 | 17 | 4.95 | 8.09 | 5.10 |
| *Periparus ater* | 3 | 0.87 | 12 | 3.50 | 6 | 1.75 | 5 | 1.46 | 1.89 | 1.13 |
| *Phoenicurus ochruros* | 1 | 0.29 | 2 | 0.58 | 1 | 0.29 | 1 | 0.29 | 0.36 | 0.15 |
| *Phylloscopus collybita* | 43 | 12.53 | 22 | 6.41 | 18 | 5.25 | 19 | 5.54 | 7.43 | 3.44 |
| *Pica pica* | 43 | 12.53 | 27 | 7.87 | 16 | 4.66 | 4 | 1.17 | 6.56 | 4.83 |
| *Picus sharpei* | 7 | 2.04 | 3 | 0.87 | 4 | 1.17 | 2 | 0.58 | 1.17 | 0.63 |
| *Prunella modularis* | 0 | 0.00 | 0 | 0.00 | 1 | 0.29 | 0 | 0.00 | 0.07 | * |
| *Psittacula krameri* | 1 | 0.29 | 1 | 0.29 | 2 | 0.58 | 0 | 0.00 | 0.29 | 0.24 |
| *Ptyonoprogne ruspestris* | 0 | 0.00 | 2 | 0.58 | 39 | 11.37 | 0 | 0.00 | 2.99 | * |
| *Serinus serinus* | 1 | 0.29 | 5 | 1.46 | 7 | 2.04 | 4 | 1.17 | 1.24 | 0.73 |
| *Sturnus unicolor* | 9 | 2.62 | 5 | 1.46 | 40 | 11.66 | 13 | 3.79 | 4.88 | 4.62 |
| *Sturnus vulgaris* | 0 | 0.00 | 0 | 0.00 | 3 | 0.87 | 3 | 0.87 | 0.44 | * |
| *Sylvia atricapilla* | 1 | 0.29 | 1 | 0.29 | 1 | 0.29 | 1 | 0.29 | 0.29 | 0.00 |
| *Sylvia melanocephala* | 1 | 0.29 | 2 | 0.58 | 1 | 0.29 | 1 | 0.29 | 0.36 | 0.15 |
| *Troglodytes troglodytes* | 2 | 0.58 | 0 | 0.00 | 1 | 0.29 | 1 | 0.29 | 0.29 | 0.24 |
| *Turdus merula* | 8 | 2.33 | 12 | 3.50 | 28 | 8.16 | 5 | 1.46 | 3.86 | 2.99 |
| Abundance/Density (birds/10 ha) | 496 | 144.56 | 564 | 164.38 | 611 | 178.08 | 298 | 86.86 | 143.47 | 40.17 |
| Richness (global values) | 29 | | 29 | | 27 | | 28 | | 28.25 | 0.83 |
| Diversity (nats) | 2.46 | | 2.46 | | 2.63 | | 1.84 | | 2.35 | 0.30 |

* Standard deviation (SD) higher than the average values ($\overline{X}$).

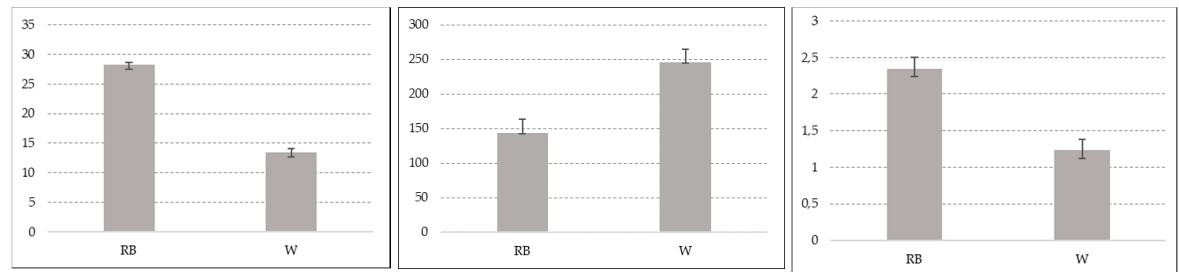

**Figure 6.** (**left**) Richness (number of species), (**middle**) Density (birds/10 ha -RB-, birds/km -W-) and (**right**) Diversity (nats) in the winter on the river Manzanares (2019–2020): RB, Riparian birds; W, Waterfowl.

## 4. Discussion

Analysis of the current regulations at a European, national and local level appears to show that there is a clear legal framework that is sufficiently broad to enable the pursuit of environmental and territorial biodiversity protection policies.

Although there is no current review available of biodiversity conservation projects in the local and regional scales in Europe, there are similar experiences in Sweden [71] or France [72–75] or, beyond Europe, the gathered by Miller et al. [74] and Stokes in the USA [75], among others.

However, the approval and publication of these regulatory frameworks is not always a guarantee of good results in terms of public policy making [76]. This is clearly observed at the state and regional level. However, among the legislative instruments that were identified here, it should be noted the

*"Royal Decree 556/2011 for the development of the Spanish Inventory of Natural Heritage and Biodiversity"* could be a very effective tool to achieve the objectives set out in this study. Under this legislative instrument, an Inventory Committee is established that is part the European environmental information and observation network, in which the regional administration in Madrid is responsible for generating the basic information necessary. However, this aspect is dealt with through the IDEM (Spatial Data Infrastructure of the Community of Madrid), which is very limited in terms of its contribution of cartography and in house databases. In addition, the Inventory indicated above should be nourished by regional catalogues and in particular, those that center on *Habitats in danger of disappearance ("Hábitats en peligro de desaparición"),* an inventory that has yet to be implemented despite its inclusion in Article 9 of the legislation "*Natural Heritage and Biodiversity Law of 2007*" ("*Ley del Patrimonio Natural y la Biodiversidad* de 2007") [77].

It would be logical if through the information provided by national and regional catalogues and inventories, local administrations were to carry out their own biodiversity conservation projects, and use this information for land-use planning and management. However, as indicated above, the situation is actually quite different and the flow of information, which should logically be top-down, is generated, albeit occasionally (18 projects in 179 municipalities), the other way round, down-top [78,79].

This is a worrying situation because the primary element of territorial transformation in Spain, and consequently of the landscape and the environment, is urbanization. In the Madrid region in the period 2005-2017, there was an increase of 4.1% in urban land (+3210.20 ha) and of 31.1% (+82,828.58 ha) in parcels of developable land suitable for immediate urbanization (Figure 7) [80]. When analyzed since 1998 [81], the increase in the evolution of urban land is considerably greater (26.59%, +17,115.90 ha), despite the significant number of empty dwellings (263,279 in 2011, 10.1% of the total) [82] (Figure 8).

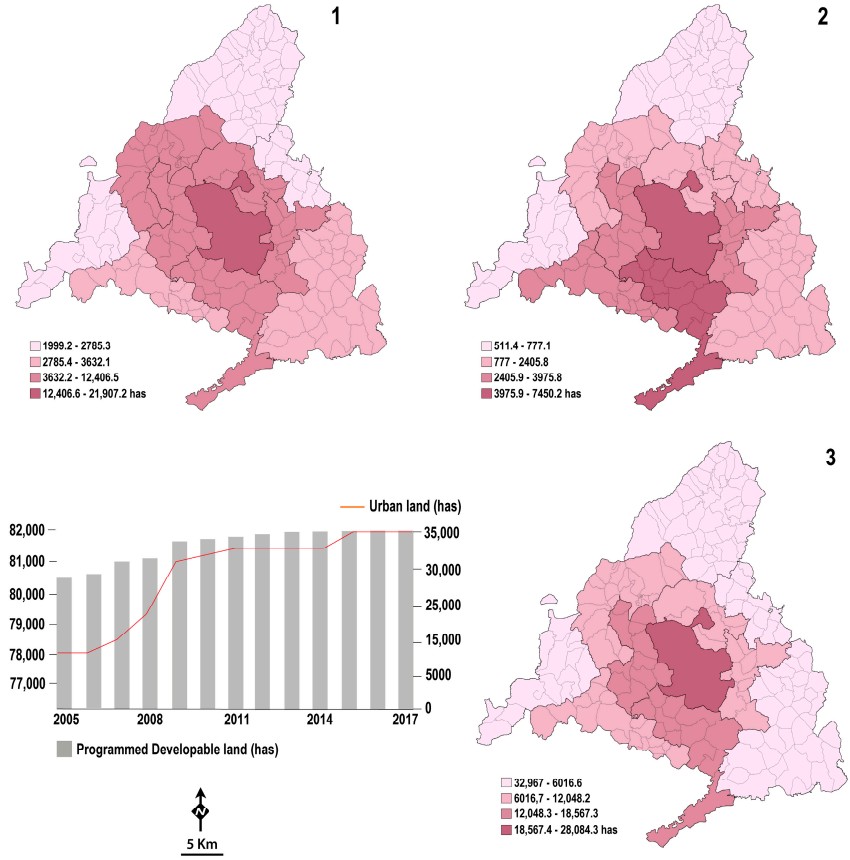

**Figure 7.** Distribution area in hectares of urban land (1), Programmed Developable land (2) and total Developable land (3) grouped in statistical areas. Evolution showing the progression of Urban and Programmed Developable land from 2005 to 2017. Data source: [28].

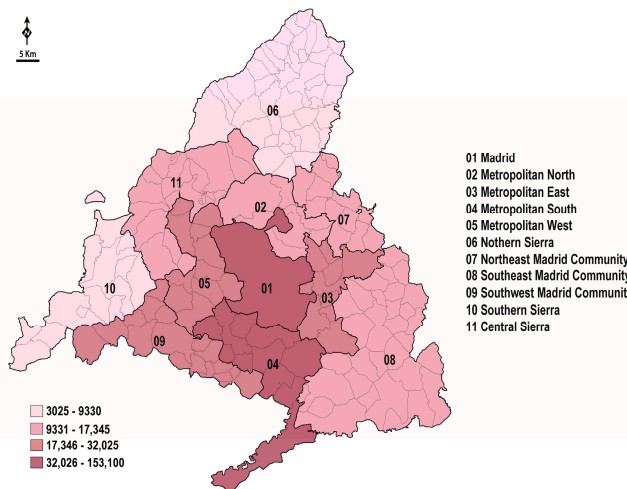

**Figure 8.** Number of empty dwellings in the Community of Madrid grouped in statistical areas. Data source: [28].

Under the current State Land Legislation [83], any public policy for land regulation, management, occupation, transformation and use should aim to: "*use this resource [land] in the common interest and according to the principle of sustainable development*" (Article 3.1.). In addition, "*the effectiveness of measures for the preservation and improvement of nature, flora and fauna, and for the protection of cultural heritage and landscape*" should also be fostered (Article 3.2.a). The Constitutional Court Judgement 61/1997, of March 20th, gives the Autonomous Communities all urban planning competences. In the case of the Community of Madrid, its Regional Land Legislation [84] specifies that the General Plans are the basic instruments for the development of municipal urban policies (Article 41.1.), in which land classification is one of the main objectives (Article 41.2.a). In addition, according to article 20 of the *Law 21/2013, of December 9th, on environmental assessment" (Ley 21/2013, de 9 de diciembre de evaluación ambiental)* [16], an environmental analysis shall also be provided. Furthermore, an assessment of the environmental impact of urban planning in Madrid must be carried out through a procedure called *Strategic Environmental Assessment*, in accordance with the provisions of the aforementioned applicable legislation (*Law 21/2013, of December 9th, on environmental assessment*). Annex IV of this legislation details the content of the strategic environmental study required for the approval of urban development plans. Point 3 states that the environmental analyses must include information on the environmental characteristics of the areas that may be significantly affected by the plan. Likewise, point 6 specifies that the plan must indicate any significant environmental impact on the biodiversity, fauna or flora, among other issues. However, this legislation does not define either the scope or the quality of the information that must be provided. As a result, strategic environmental assessment is often undertaken based on poor quality information that often does not reflect the global values, or the stresses and environmental protection needed for the territory. In addition, the promoters of these plans are themselves responsible for providing the environmental studies required for their final approval, which undoubtedly results in a clear conflict of interest.

Therefore, as tools for environmental conservation, the General Urban Development Plans are underused urban development planning instruments, despite their varied functional possibilities. The information that they should incorporate and use to protect the biodiversity in the territories they deal with is generally not available. Moreover, information relevant for environmental assessment and for urban land planning, either developable or protected, is not explicitly generated. Thus, from an environmental point of view, few concrete instruments/plans are drawn up, and they often do not reflect the actual situation of the flora, fauna or habitat. This has often enabled urban developments to be promoted in areas of great environmental value that were previously classified as undeveloped

land. As in some other case studies, the type of policy implemented and the process of formulating these plans is of great importance in order for them to be successful [85,86].

The case of Torrelodones represents a frequent situation in the Madrid region, a clear example of what has been indicated above. As such, 53.17% of the surface area of this small municipality is included in two protected natural spaces, the 'Cuenca Alta del Manzanares' Regional Park (PRCAM) and the 'Cuenca Media del Río Guadarrama' Regional Park (PRCMG) [87], both Special Areas of Conservation (SACs) integrated within the Natura 2000 network. The PRCAM was created in 1985 under the *Law 1/1985, of January 23rd, creating the Regional Park of the Cuenca Alta del Manzanares ("Ley 1/1985, de 23 de enero, de creación del Parque Regional de la Cuenca Alta del Manzanares")*. For its part, the PRCMG was created by virtue of the *Law 20/1999, of May 3rd, on the Regional Park of the Middle Course of the Guadarrama River and its surroundings ("Ley 20/1999, de 3 de mayo, del Parque Regional del Curso Medio del río Guadarrama y su entorno")*. In some cases, the Urban Land in Torrelodones borders the strictly protected areas of these two parks (Integral Biological Reserve) [87]. However, in regards to the undoubted value of the municipal territory, specific data on local biodiversity had not begun to be collected until 2018, that is 33 and 21 years after the creation of the two natural spaces in which a good part of the municipality is located. These tasks have been carried out by the local administration, despite the regional administration being the one that holds virtually all of the ability to address environmental matters. Even in 2020, Torrelodones is still the only municipality in Madrid that has municipal information of this nature, and at this level of detail. This has made it possible to specify the environmental relevance of the municipal territory, affected by strong territorial tensions and urban pressures due to its location in the peri-urban area of Madrid.

Indeed, the value of the vascular flora of Torrelodones is particularly high, contributed to by 767 types of vascular plants. A comparison of the number of taxa identified in the 21.9 km$^2$ over which the municipal terminus extends with the records of other countries and areas in Europe corroborates this assertion: 1521 species in Fennoscandia (1,288,125 km$^2$ in Denmark, Norway, Sweden and Finland) [88] (Sætersdal et al., 1998); 2922 indigenous and neophytic taxa in areas of high diversity like the Austrian Alps (53,500 km$^2$) [89]; 1423 species in Belgium (32,592 km$^2$) (Stieperaere, 1979) [90]; 2049 species in the British Isles (315,134 km$^2$) [91]; 3556 in the Czech Republic (78,866 km$^2$) [92]; 3660 species in Germany (357,386 km$^2$) [93]; 6276 species including 739 non-native elements in the Iberian Peninsula (583,832 km$^2$) [91]. Some 1 km$^2$ UTM grids in Torrelodones house more than 300 species. In particular, grid 30TVK2291 (362 taxa) has 13.76% of the regional flora in 0.01% of the regional surface area and 5.77% of the Iberian flora in 0.004% of the peninsular surface area. Indeed, a Review of the Urban Regulations of this environmentally valuable space, referred to as the *Northern Homogeneous Area* in local planning [94], attempted to modify the land classification to embark on its urban development, installing 700–1200 homes together with a golf course and other services on this semi-natural area of great environmental value. This development was justified by the city council at the time on the grounds of the little value of the area and if it had gone ahead, it would have led to the disappearance of one of the areas of greatest biodiversity in the center of the Iberian Peninsula.

The lack of specific georeferenced information at different levels is one of the main problems for the proper management of these territories, both in terms of the sustainability and conservation of the biodiversity at these sites. Quality studies of these territories are necessary, with precise data of this nature. A good example of this type of study is the series of articles *Database for Biological Flora of the British Isles* published in the Journal of Ecology since 1941, which amongst other information provides a georeferenced database of 353 species of the British flora [95]. Along similar lines, the *Flora on* project [96] in mainland Portugal is also of particular interest, the objective of which is to collect and make available to the public chorological, ecological, morphological and photographic information on the vascular flora of Portugal through the platform http://flora-on.pt/. Other projects with georeferenced information are also of note, including the *Database BiolFlor* project in Germany, an information system on vascular plants [93], and the *PLADIAS* project [97] in the Czech Republic [98]. Studies of this kind, with georeferenced information on the distribution of species, are an essential

and very useful tool for adequate territory management in terms of environmental sustainability. Moreover, specific interventions are needed to preserve biodiversity, developed within the framework of municipal competences at the local level.

The plan for the re-naturalization of the river Manzanares, a specific action implemented by the Madrid City Council between 2015 and 2019, highlights the environmental benefits of well planned and executed municipal actions. Until these activities to recover the river and its banks were undertaken, the river in the area analyzed was a channel with a concrete base, and the canalization and breakwater banks were created after the modifications carried out between 1945–1962, after the channel was established between 1914–1925 [99–101]. The river, without vegetation on the riverbed or banks, was segmented by nine small dams that formed a succession of still water. The renaturalization commenced by opening the gates of the aforementioned dams to allow the river to flow freely, mobilizing the sediments and encouraging the subsequent growth and colonization of vegetation.

Quantitative data regarding the populations of waterfowl and riparian birds prior to the renaturalization of the river are not available due to the limited interest in this group of birds over this stretch of the river. There is, however, information available from an adjacent stretch of the river located downstream, which remained in acceptable natural conditions over time [102]. It is interesting to note that the densities of these species recorded in the area recovered exceed those observed downstream in areas with little human intervention. Both the global data and the density values of the dominant species like *Anas platyrnhyncos, Gallinula chloropus, Cettia cetti, Passer montanus* or *Turdus merula* demonstrate increases.

Perhaps one of the most remarkable facets of the renaturalization of the River Manzanares are that the improvements observed were achieved on a relatively small budget. Indeed, the overall cost of execution of this plan (€1,216,054) and the weighted cost per kilometer of river (€162,140.53) are far below the cost of other seemingly comparable projects carried out on urban river banks located in the same territorial context: €978,420.34/km of the Henares river in Guadalajara (Guadalajara), €2,108,107.25/km of the Tagus river in Toledo (Toledo) or €1,002,378.57/km of the Tagus river in Talavera de la Reina (Toledo) [103–105]. The cost of this *Plan* is also far below the budget for another project undertaken in the same area, that of *Madrid Río*, with the renaturalization of the Manzanares representing less than 0.5% of the cost of this second plan, which amounted to €420 million [106]. Likewise, the river Manzanares plan cost only 0.03% of that of the renovation works on the M-30 motorway that ran partly adjacent to this stretch of the river and that is thought to have cost €3600 million, although according to other sources the final cost of this project was more likely to be between €5600 and 7000 million [107].

The aforementioned Madrid Río project deserves special criticism. Its development was linked to the renovation of the urban M-30 motorway between 2003 and 2011 [108], a few years prior to the renaturalization of the river Manzanares in the same space. Together, these interventions have been considered as "the most important work carried out in the city of Madrid in recent decades and probably, one of the most ambitious projects in a public space recently implemented in Europe" [106], and they offer us a prime example of how the recovery and integration of urban river spaces is often pursued under a gross misconception. Indeed, no part of the budget for this high-cost project (€420 million) was allocated to the recovery, restoration or renaturalization of the river Manzanares, while tens of million euros were devoted to the landscaping of its banks.

## 5. Conclusions

European legislation on environmental matters currently provides us with an optimal reference framework to program suitable of actions on a national and regional level. Unfortunately, the strategies of Autonomous Communities and States regarding the conservation of biodiversity often does not advance beyond the theoretical context, not reaching a practical level on the ground. The legal framework within Spain allows effective public policies to be implemented in order to protect nature, supported by the mandatory biodiversity inventories that ought to be maintained according to current legislation. However, and in addition to having a limited scope in terms of their content, these databases

do not provide the local information necessary to act on specific territories. Since the information available is clearly insufficient, in Spain the impact of territorial transformation (i.e., the process of urbanization) on the main elements, and consequently on the landscape and biodiversity, cannot be effectively analyzed and evaluated from an environmental perspective. This is an issue that falls within the competence of local development but that is regulated and endorsed by the regional administrations. It is precisely these latter organisms that hold all the competences governing environmental protection, leaving local initiatives very little room in which to maneuver.

In the case of the Autonomous Community of Madrid, the region with the highest population density in Spain, the public policies that most intensely affect the landscape and environment are designed based on very limited knowledge of the existing environmental values and stress. The limited information available does not favor accurate diagnoses, from which plans, programs or projects can be developed in line with the existing principles of environmental sustainability and preservation of biodiversity. This circumstance has often caused a significant increase in the environmental deterioration of the regional territories, mainly as a consequence of increasing the urban land and that which can be developed in areas of great natural value. It is therefore essential to increase the quality and quantity of the information available on biodiversity at a local level, as without more detailed knowledge the territories controlled on a local level cannot be properly managed.

A good example of the above is the case of Torrelodones, where in 2005 an attempt was made in the municipality to reclassify undeveloped land that would have destroyed one of the most valuable areas of biodiversity in the region and indeed, in central Spain. The vascular flora in that area had tremendous value, even in terms of the landscape of the Iberian Peninsula, the true value of which was not known until 2019 thanks precisely to a municipal initiative aimed at gaining a better understanding of the local flora. This example makes it clear that studies of this kind should be promoted by the regional administration given that it is the body that holds the appropriate competencies. It is they that should provide local entities with optimal instruments to act appropriately on a territory, provided that there is the political will to do so. In the absence of studies of this nature conducted by the regional administration, municipal initiatives represent possible alternatives to improve the quality of the environmental information that is essential to be able to sustainable manage municipal territories. The economic costs of local studies focused on biodiversity are minimal and they represent a reasonable proposition for any municipal administration, given the intense benefits that arise from enhancing this patrimony.

Improving the quality and the level of detail of the available environmental information is the first step in developing territorial impact plans and global or specific projects that actively contribute to the conservation of biodiversity. The *Plan for the renaturalization of the river Manzanares as it passes through the city of Madrid* is an excellent example of how a local initiative can contribute to preserving and improving biodiversity within an urban space, even though this is a densely populated area that has suffered intense human intervention.

**Author Contributions:** P.M.-H., N.L.-E. and A.-B.B.-M. devised the idea for this study, collected and analyzed the data, and wrote the original manuscript; F.A.-Á. assisted in the data collection and analysis, generated the figures included in the manuscripts and reviewed the manuscript and the associated literature; M.d.C.-T. assisted in the data collection and reviewed the manuscript. All authors have read and agreed to the published version of the manuscript.

**Funding:** This research was funded by Erasmus+ Project RailtoLand (Grant agreement: 2019-1-ES01-KA203-065554—Erasmus + Program of Europe Union).

**Conflicts of Interest:** The authors have no conflict of interest to declare.

## Appendix A

**Table A1.** Habitat of Annex I of the Habitat Directive and communities in Torrelodones (Madrid). EUc: Habitat Code of the European Union, (*) Priority Habitats. Source: [41,49,56].

| Communities of Slopes and Rocky Areas |
| --- |

- *Quercus rotundifolia* Forests: *Junipero oxycedri-Quercetum rotundifoliae* Rivas-Martínez 1965 (EUc 9340. *Quercus ilex* and *Quercus rotundifolia* forests)
- Juniper Forests: *Junipero* oxycedri-*Quercetum rotundifoliae* Rivas-Martínez 1965), (EUc 5210. Arborescent matorral with *Juniperus* spp.)
- *Cytiso scoparii-Retametum sphaerocarpae* Rivas-Martínez ex Fuente 1986 (EUc 5330 Thermo-Mediterranean and pre-desert scrub)
- *Rosmarino-Cistetum ladaniferi* Rivas-Martínez 1968 (EUc 4030 European dry heaths)
- *Festuco amplae-Agrostietum castellanae* Rivas-Martínez & Belmonte 1986/*Gaudinio fragilis-Agrostietum castellanae* Rivas-Martínez & Belmonte 1986 (EUc 6200 Semi-natural dry grasslands and scrubland facies) (*)
- *Poo bulbosae-Trifolietum subterranei* Rivas Goday 1964 (EUc 6220. Pseudo-steppe with grasses and annuals of the *Thero-Brachypodietea*)
- *Asplenio billotii-Cheilanthetum hispanicae* Rivas Goday in Sáenz & Rivas-Martínez, 1979 (EUc 8220. Siliceous rocky slopes with chasmophytic vegetation)
- *Evaco carpetanae-Sedetum andegavensis* Rivas-Martínez, Fernández-González & Sánchez-Mata, 1986 (EUc 8230. Siliceous rock with pioneer vegetation of the *Sedo-Scleranthion* or of the *Sedo albi-Veronicion dillenii*)

| Communities of Valley Floor, Wetlands and Riverbanks |
| --- |

- *Ficario ranunculoidis-Fraxinetum angustifoliae Rivas-Martínez & Costa in Rivas-Martínez, Costa, Castroviejo & E. Valdés 1980/Salici salviifoliae-Fraxinetum angustifoliae Lara et al., 1996* (EUc 91B0. Thermophilous *Fraxinus angustifolia* woods)
- Tree schrub *of Salix salviifolia: Salicetum salviifoliae* Oberdorfer & Tüxen in Tüxen & Oberdorfer 1958 [*Salicetum salviifolio-purpureae* Rivas-Martínez 1965 (art. 43, syntax. syn.), *Salicetum lambertiano-salviifoliae* Rivas-Martínez 1965 corr. Rivas-Martínez, Fernández-González & Sánchez-Mata 1986 (syntax. syn.)] (EUc 92A0. *Salix alba* and *Populus alba* galleries)
- **Riparian forests of** *Populus alba: Populion albae* Br.-Bl. ex Tchou 1948 (Euc 92A0. *Salix alba* and *Populus alba* galleries)
- *Trifolio resupinati-Holoschoenetum* Rivas Goday 1964 (EUc 6420. Mediterranean tall humid grasslands of the *Molinio-Holoschoenion*)
- *Junco pygmaei-Illecebretum verticillati* (Bellot 1953) Tüxen 1958 (EUc 3170. Mediterranean temporary ponds) (*)
- *Sedetum lagascae* Rivas-Martínez, Fernández González, Sánchez-Mata & Sardinero in Rivas-Martínez, T.E. Díaz, Fernández González, Izco, Loidi, Lousã & Penas 2002 (EUc 3170. Mediterranean temporary ponds) (*)
- *Callitricho brutiae-Ranunculetum peltati* Pizarro & Rivas-Martínez 2002 (EUc 3150. Natural eutrophic lakes with Magnopotamion or Hydrocharition -type vegetation)
- *Callitricho stagnalis-Ranunculetum saniculifolii* Galán 1999 (EUc. 3150. Natural eutrophic lakes with *Magnopotamion* or *Hydrocharition* -type vegetation)
- *Lemnion minoris* Tüxen ex O. Bolòs & Masclans 1955 (EUc. 3150. Natural eutrophic lakes with *Magnopotamion* or *Hydrocharition* -type vegetation)
- *Paspalum paspalodes: Ranunculo scelerati-Paspaletum paspalodis* Rivas Goday 1964 corr. Peinado, Bartolomé, Martínez-Parras & Olalla 1988 (EUc 3280. Constantly flowing Mediterranean rivers with *Paspalo-Agrostidion* species and hanging curtains of *Salix* and *Populus alba*)

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
