# Peer review of "The Contribution of Local Management to Biodiversity Conservation: An Analysis of Specific Cases in the Region of Madrid (Spain)"

_land, doi:10.3390/land9110462_

Round 1
Reviewer 1 Report
It is a work oriented fundamentally to the legislative aspects on biodiversity.
Introduction
We suggest expanding the existing references on floristic diversity and plant communities in the study areas (Torrelodones, Manzanares river).
Material and methods
The areas chosen for the study are not comparable, they have very different sizes, and the urban influence is very different.
In the renaturation of the Manzanares, there is no mention of the species used, if it is a renaturation they should be native plants, but the authors say nothing about it. They must provide a list of plant species.
The botanical sampling method used is not clear to me, rather it seems that they have used previously published listings.
It is not clear how the study of flora and vegetation has been carried out, and if the autochthonous flora has been separated from the introduced flora, the authors should provide a table with the autochthonous flora and the one introduced in each of the study areas. it would mark the urban influence in the study carried out, and especially in the type of existing birds.
Some paragraph of methodology could go to results.
Results
The authors say that the genus Ortegia is endemic to the Iberian Peninsula, I suggest you see Flora Ibérica, you must specify its area of distribution.
Line 296 .., Says, 19 types of habitats identified ..., "the authors must provide a table with the 19 types of habitats, including the names of the plant associations"
Line 316 the authors speak of Ophioflossum azoricum C. Presl., Flora Ibérica does not give it in Madrid, I suggest they confirm it, there may be errors, they can confuse it with Ophioglossum lusitanicum L.
Author Response
Dear Reviewer,
We would like to thank you for the time and effort that you have put into reviewing our manuscript and we are grateful for your comments, all of which we feel are relevant and of value. Indeed, we believe that in addressing these issues we have improved our work considerably. We hope that in the light of our responses to the issues raised now you will find our article suitable for publication.
Many Thanks,
The authors

Reviewer 2 Report
Abstract: it would be good to add the country of the study area to the abstract.
Introduction is sound and coherent with clear research objectives.
Methods: how were the criteria defined? How was the qualitative assessment done?
Results are properly presented.
Discussion: this section lacks the comparisons between the findings of this study and similar studies conducted in other countries. Moreover, little discussion has been provided around the implications of the findings for researches and authorities of other countries who read the paper which need to be expanded.
Author Response

(The authors gave the same response as above.)

Reviewer 3 Report
The article investigates the role of the local administrations in improving the biodiversity database through two cases.
The article has been well organized, with a clear statement of the question, sufficient introduction of background, an adequate description of the methods, and results.
I just have one suggestion. It will be better if the authors could add a few sentences in the abstract to elaborate on the results and findings.
Author Response

(The authors gave the same response as above.)
